# A Practice for the Design of Interactive Multimedia Experiences Based on Gamification: A Case Study in Elementary Education

Carlos Alberto Peláez and Andrés Solano *

Faculty of Engineering, Universidad Autónoma de Occidente, Cali 760030, Colombia
* Correspondence: afsolano@uao.edu.co

**Abstract:** This research proposes a practice for the design of interactive multimedia experiences, based on gamification and applied in elementary education. The practice is expressed in the Essence graphic notation language. Through practice, a multimedia experience based on gamification, called Coco-Shapes, was developed to assist learning of the English language in the following topics: geometric figures, colors and counting. The study is aimed at children aged between 4 and 5 years of age, in the transition grade in a private school, and in a vulnerable community with scarce resources in the city of Cali (Colombia). A research process was conducted for this experience, with the participation of two groups of students: one experimental group and one control group. The results are optimistic, since it is evident that, through carrying out the activities that make up the practice, the solution achieved contributes to increased user learning, as well as favors a greater receptivity in the students towards the use of technology in the training process.

**Keywords:** Interactive Multimedia Experience (IME); elementary education; gamification





## 1. Introduction

A learning experience refers to any interaction, course, program, or other experience in which learning takes place. It can occur in traditional academic settings, such as schools and their classrooms, or in non-traditional, out-of-school, and outdoor settings. Similarly, it can be sustained through traditional educational interactions, or non-traditional interactions, where the student learns through games and interactive software applications [1].

On the other hand, a "multimedia system allows for value creation for interested parties through the deployment of an Interactive Multimedia Experience (IME), using an ethic and responsible design approach, and addresses the users' needs, interests and expectations by influencing their human senses via storytelling using digital media resources" [2]. The IME is considered to be a key element in the design of a multimedia system, through which value is delivered to its stakeholders [2]. In a review of the state of the art methodologies, related to the development of interactive and multimedia systems [3], it was clear that no work approach is concerned with establishing a precise and formally defined process to guide the design of a multimedia system.

A definition of the concept of 'Gamification', consistent with this research, is "the use of game design elements in non-game contexts" [4]. Different researchers conclude that gamification in learning has a massive appeal among students by stimulating motivation, commitment, and social influence [5–7]. However, despite the fact that there are methodological approaches to designing gamified learning experiences [8,9], careful design is required for the experiences to be effective. Taking the above into account, in this article, we propose a practice for the design of IME based on gamification elements [10], such as goals and objectives, rules, narrative, freedom to choose, freedom to make mistakes, rewards, feedback, visible status, cooperation and competition, time constraint, progress, and surprise. The practice is expressed in the Essence graphic notation language, and this was applied in the context of elementary education for its preliminary validation.

One interesting factor is related to the opinions and valuations made by users in the Google and Apple stores, regarding these kinds of solutions. Although there are applications that aim to improve the academic performance of students, there are limitations and evidence of unfavorable comments regarding their use [11]. This highlights the importance of analyzing the impact of the design process of gamification-based solutions and the need to consider factors that influence the success of a learning experience based on gamification elements in an educational context [7]. Some of these factors are related to the way in which a multimedia experience based on gamification can contribute to an improvement in the learning of the user, who is the object of the learning process. They also relate to the way in which the multimedia experience can contribute to the favorable attitude of the participant towards the technology, which allows a significant prediction of their reaction.

The design process of the IME plays a crucial role in the inclusion of these factors in the specificity of the kind of solution that is the object of production, considering an effective delivery of value to the stakeholders, according to the purpose and context of its use. However, the existing restrictions are clear, due to the limitation of methodological approaches that define a guiding process for the design of an IME.

This research presents a practice for the design of IME and its use in the development of interactive multimedia experiences for English language learning, called Coco-Shapes, and is based on gamification elements. The development of this experience is aimed at children between four and five years of age, who are attending a private school located in commune 20, in the city of Cali, Colombia [12]. This commune is characterized as being one of the most socially and economically vulnerable in the city, according to the Social Observatory of the Mayor's Office of Santiago de Cali [13,14].

It should be noted that, in the school context of elementary education in Colombia, the teaching of a second language, such as English, is mainly restricted to the higher socio-economic strata of the population [15]. This restriction is due to the high costs for the educational institutions to provide an academic offer that promotes bilingualism throughout the student's elementary education cycle. Given the social and economic scenario outlined above, one of the main constraints that had to be considered during the design process of the multimedia system was the need to develop a low-cost solution.

Through our research, we provide a new approach in the development of IME for elementary education, in public or private institutions with scarce resources, contributing to the development of learning experiences in a school context. This approach is presented by means of an artifact [16], which is represented through a practice for the design of IME using the Essence graphic notation language [17]. The case study presented here is guided by a process for the development of an IME based on gamification for English language learning, aimed at children in transition, between 4 and 5 years old.

The practice for the design of an IME has been adapted so that it can be applied by companies that create technology-mediated solutions and content for educational contexts. Likewise, it is aimed at technology support units, or Information and Communication Technology (ICT) managers in schools, as a resource for teachers who use technology to develop their teaching/learning strategies in their subjects.

This article is organized as follows: Section 2 describes the materials and methods applied in this research. Section 3 presents the research results. Section 4 presents discussion regarding the results and, finally, Section 5 presents the conclusions and future work.

## 2. Materials and Methods

### 2.1. Creation of Work Teams

The project for the development of IME in elementary education began in July 2021, at the Universidad Autónoma de Occidente in the city of Cali, Colombia. This project involved a core group with a total of five researchers: two researchers with a PhD in Engineering, one researcher with a PhD in Education, and two designers with a master's degree in Design and Interaction. These researchers belong to research groups recognized

by the Ministry of Science and Technology of Colombia, Minciencias [18]. The second group consisted of five interns: three professionals in the field of Multimedia Engineering and two in Graphic Design. Both of these groups make up the work team responsible for carrying out the project.

### 2.2. Description of Key Stakeholders

The IME, called Coco-Shapes, is aimed at students in the transition grade of a school in the city of Cali, which is committed to an educational strategy of bilingualism for low-income people [19]. The total number of transition students in 2022 was 36 (children between 4 and 5 years old). Throughout the process, the students were accompanied by two teachers, who are pedagogues and experts in the field of English learning, as well as the school principal, who has a degree in foreign languages.

Once the regulations and laws that influence the design of the Coco-Shapes were identified (related to data processing, type of audiovisual content, materials of physical objects, etc.), the work team reviewed the adequate implementation in the prototypes obtained, with the support of a leading NGO (Non-Governmental Organization), for the design of interactive experiences in the educational sector [20].

### 2.3. Research Methodology for Designing the Solution

The methodology used to guide the research was based on the principles of design science research methodology [21]. The approach suggests a set of guidelines for its application [16], and this paper discusses the following: (i) the basis of a relevant problem, which, in this case, it is the limitation of existing methodologies for the development of interactive and multimedia systems, specifying a working approach for the design of interactive multimedia experiences; (ii) a practice for the design of IME, which is the a solution artifact, (iii) the evaluation of the practice, which is a case study for the school context, and (iv) a dissertation on the contribution and limitations, which is an evaluation of the practice for the design of IME.

Based on the above, as a validation mechanism for the case study, the following hypotheses were defined. This study proposes a set of hypotheses based on the results discussed in previous studies [6,7], related to the contribution offered by gamification to the learning process of students. In this case, our study consists of knowing the contribution of a multimedia experience, as part of a gamification strategy, to the learning experience of students in elementary education. In addition, we want to know about the emotional behavior of the user when interacting with a multimedia experience that includes gamification elements, as other authors have explored [22].

**Hypothesis 1 (H1).** *A multimedia experience based on gamification elements offers a contribution to the increased learning of the user, who is the object of the learning process.*

**Hypothesis 2 (H2).** *A multimedia experience based on gamification elements contributes to the favorable attitude of the participant towards the technology, allowing the significant prediction of their reaction.*

### 2.4. Practice for the Design of IME in Elementary Education

For the design of the practice, as an artifact of the solution, the graphical notation language, offered by the Essence standard of software engineering [17], was used. Essence is a simple model of the challenges that all software development teams face, coupled with a visual language to capture practices to help teams address those challenges. The model contains practices, alphas, activities, roles, etc. [17]. Essence enables practices and related knowledge to be expressed in a simple, visual way that ensures that they can be easily shared, understood, adopted, adapted and applied both independently and in combination with other practices (for example the Minimum Viable Multimedia System practice [23]).

Figure 1 shows the adaptation of the practice for the design of Interactive Multimedia Experiences (IME) [24] in the context of elementary education.

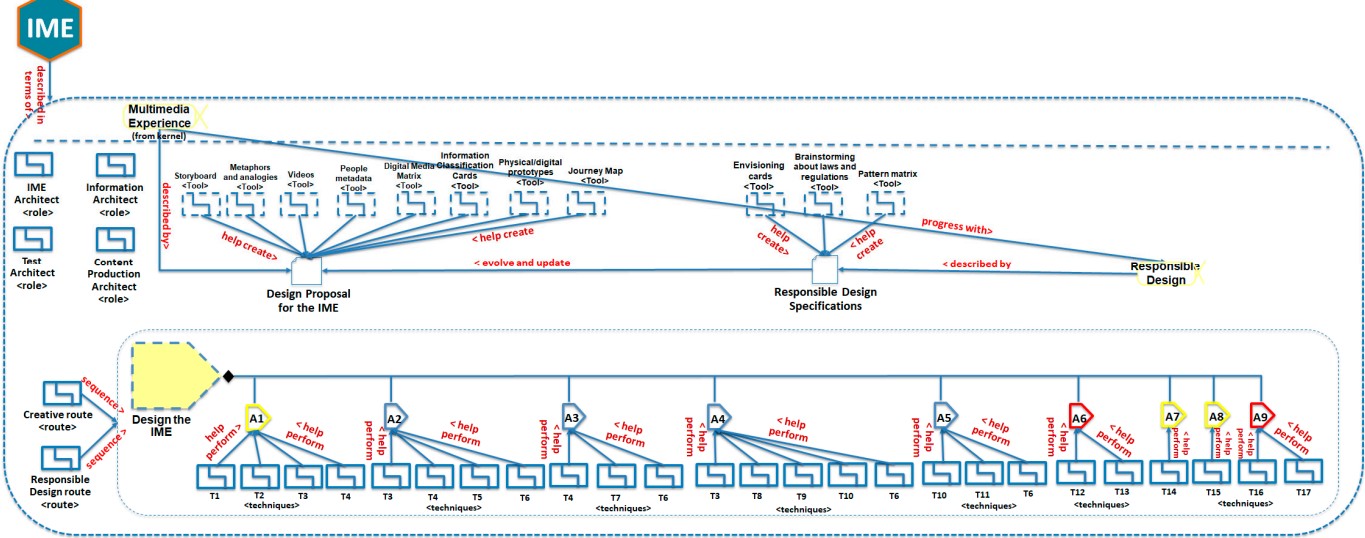

**Figure 1.** Practice for the design of IME in elementary education.

This practice is composed of a set of activities, techniques, tools and Alphas to guide the design process of the IME in elementary education. For the organization of the flow of activities that guide the work teams, two work routes were defined [24]: the first is the Creative Route and the second is the Responsible Route. The work routes suggest a certain degree of sequentiality the first time they are carried out. However, these can be conducted concurrently among the other routes that exist for the pre-production of multimedia systems [2]. This implies the challenge of knowing the status of completion of each of the routes. The routes were proposed considering a model focused on concurrent engineering [25], where a set of states was defined for each route; this allows a work team, through a checklist, to know the current state of the project and report it to the stakeholders.

### 2.4.1. Creative Route

The Creative Route includes activities that allow the conception of the design of the IME to be developed and that allow a preliminary estimate of what is required to achieve its implementation. The objectives of this route are the following:

- To define the different elements that structure a story; these respond to the interests of the base problem identified, as well as to the learning objectives being studied.
- To define the foundations, as well as the structure of the design of the multimedia experience, focusing on the interests, needs and expectations of students and teachers.
- To produce a preliminary definition of the digital media, necessary physical environments, sensory perceptions, interaction patterns, information sources, gamification elements and technologies that will potentially be involved in the design of the IME.
- To perform the tests focused on the conception of the multimedia experience conceived.

This route is called 'Creative', because it describes the process flow that connects the different activities and techniques, oriented to the design of a multimedia experience that allows the learning objectives to be achieved; this involves a story that is based on the definition of a problem. These elements are the pillars on which the conception of an IME is produced, where the digital media, the necessary physical environments, the sensory perceptions, the interaction patterns and the different sources and processes of information transformation (which generate value for the user, before, during and after the experience), are preliminarily identified.

In order to carry out the Creative Route, it is essential to count on the active and collaborative participation of the following stakeholders, from a elementary education context:

- Pedagogical advisor: accompanies the work team to articulate the educational guidelines, competencies under study, game mechanics and gamification that drives the project in the multimedia experience. They also participate in the conceptualization, concretion, and specification of the interactive multimedia experience, by defining its narrative line and visual identity.
- ICT Manager: facilitates active communication within the work team that will implement the interactive experience with the stakeholders of the educational institutions. In addition, it promotes the active participation of teachers and students in the processes of conception, concretion and specification of the elements that make up the multimedia experience.
- Teacher: as an expert in the discipline where the problem under study is located, they can actively collaborate with the pedagogical advisor for the definition of the interactive multimedia experience, in such a way that the solution contributes to the teaching/learning process, through a series of well-defined activities.
- Students: as potential users of IME, they participate in the process of inquiry and subsequent conception of the IME, which will allow them to achieve the learning objectives under study. The participation of end users is vital to incrementally test the multimedia experience.
- Work team: refers to professionals from different fields of knowledge who participate in the pre-production of the multimedia experience. The team is responsible for designing the IME, as well as defining the information structure, the basis for the subsequent design, and the integration of the hardware and software components of the solution. In addition, the team includes people responsible for producing the multimedia content that the experience must offer.

Table 1 presents the set of activities involved in the Creative Route to achieve the scope of the work. In addition, Table 2 presents the series of steps carried out for the implementation of the technique and the tools used by the work team. The basis of these techniques, and the steps for their application, are based on approaches and frameworks related to tools used in web ethnography [20], practices based on Design Thinking [26], storytelling [27] and methods for information architecture design [28].

**Table 1.** Creative Route activities.

| Activity Code | Activities |
|---|---|
| A1 | Design of the structure and flow (narrative, temporal, events) of the story to be developed from the problem to be solved, describing the events it narrates, the characters involved, the time in which it develops and the space in which these events take place. |
| A2 | Definition of the foundations that underlie the design of an IME based on the story and the problem(s) identified, specifying the milestones of the story where the multimedia system must produce a cognitive, emotional and sensory influence on the user. |
| A3 | Definition of the types of digital media from which the production of multimedia content will be carried out, as well as the estimated physical environment settings necessary to engage the user in the continuum of the experience, before, during, and after interacting with the multimedia content. |
| A4 | Identification and classification of the different sources and processes of information transformation that occur because of the interactive multimedia experience, before, during and after the user's interaction with the solution. |
| A5 | Conception of the different sensory perceptions, as well as the different styles of interaction, guided by the design of the IME and the types of digital media necessary to ensure the psychological, cognitive, and sensory influence on the user. |
| A6 | Preliminary verification of the design conceived for the IME. |

**Table 2.** Detail of the techniques associated with the Creative Route in elementary education.

| Activity | Code Technique | Technique | Steps for Its Application | Tools |
|---|---|---|---|---|
| A1 | T1 | Video as a support for ethnography | - Define what is to be filmed.<br>- Determine who will carry out the filming.<br>- Make the necessary arrangements to obtain filming permits.<br>- Make the video.<br>- Analyze and collect information from the video. | Devices that allow video recording, according to the capture needs determined by the work team. |
| A2 | T2 | Person definitions | - Generate a list of potential users and their relevant attributes around the multimedia experience.<br>- Define a finite and controllable number of user types.<br>- Create persons associated with the user types, associating them with the previously defined attributes (name, age, gender, profession, tastes, interests, etc.).<br>- Build a visual profile for each person that is highly visual and quick to read, associating their attributes, anecdotes and quotes with each one. | Design of a format to store the metadata of the personal profiles. Tools for capturing or producing the visual elements of the format. |
| A1, A2, A4 | T3 | Metaphor and analogy production | - Determine the starting point for the use of metaphors and analogies.<br>- Identify the best metaphors and analogies for story development and implementation in the experience.<br>  \* Direct analogy (real objects).<br>  \* Fantasy analogy (objects that do not exist but may be imaginable).<br>  \* Symbolic analogy (compare aspects of the concept, with aspects of a different one).<br>  \* Personal analogy (relate oneself to the concept by placing oneself in a certain situation).<br>- Generate concepts for each metaphor and analogy produced.<br>- Document, discuss and improve the concepts. Discuss how they should be evaluated and developed in the future.<br><br>Retain low-fidelity prototypes for the metaphors and analogies produced. | - Formats with sketches that facilitate the design of metaphors and analogies. |
| A1, A2, A3 | T4 | Storyboard production | - Clearly define the solution to be illustrated.<br>- Create the characters and describe their experiences.<br>- Trace the journey through imagined situations and define points on the map where the user will encounter multimedia content, learning activities, sensory perceptions, and options for interaction with the experience.<br>- Illustrate the scenarios with a frame-by-frame storyboard, developing the narratives in each, using sketches.<br>- Share the story with other stakeholders and use their feedback to refine concepts and other aspects, such as the type of multimedia content to be used, as well as the insights, achievement of learning activities or interaction possibilities that they wish to bring to each of the points on the map.<br>- Retain low-fidelity prototypes of the storyboard produced. | Tools for sketch generation. |

**Table 2.** *Cont.*

| Activity | Code Technique | Technique | Steps for Its Application | Tools |
|---|---|---|---|---|
| A2 | T5 | Map of compelling experiences | - Select the narrated experience you wish to analyze and create a worksheet that references the following aspects: attraction, entry, engagement, exit, and extension.<br>- Describe the attraction stage, thinking about the previous interactions that can generate interest. Describe all the actions you can take for this purpose.<br>- Describe the entry stage, considering what happens when the user is brought into the experience and how this influences the end purpose.<br>- Describe the engagement stage, which is the core of the experience for the user. At this stage, do not lose sight of the learning objectives and activities, as well as the design of the assessment of those learning objectives.<br>- Describe the exit stage when the user prepares to withdraw from the experience.<br>- Describe the extension stage, specifying everything that must happen to the user after the experience to keep them engaged.<br>- Analyze each of these stages, cross check it with each of the following attributes:<br>  • Defined: Can you describe it? Is it defined?<br>  • Fresh: Is it novel, scary, surprising, amusing?<br>  • Immersive: Can it be felt, perceived? Can you lose yourself in it?<br>  • Accessible: Can you try it? Can you get it to do what you want it to do?<br>  • Meaningful: Does it make you remember, connect, think, grow?<br>  • Transformative: Does it feel different? Do you have anything out of this to display?<br>- Rate each of the six attributes of the experience in each of the five stages, representing it by means of a line that crosses the different stages and that can change thickness or color. Thickness can signify a greater degree of presence of the attribute in a stage. Color can differentiate the appreciation of different participants.<br>- Analyze the experience map and review why each of the stages evidences each outcome for the experience under analysis. Evaluate where you can improve and strengthen the experience proposition. | Space and resources available as a surface and elements for writing and working in a collaborative space. |

**Table 2.** *Cont.*

| Activity | Code Technique | Technique | Steps for Its Application | Tools |
|---|---|---|---|---|
| A2, A3, A4, A5 | T6 | Journey Map design for the experience | - Generate a list of all the specific activities and learning activities that the user will perform in the experience.<br>- Relate the specific activities to the learning objectives within the experience.<br>- Generate a list of digital media and associate their display with each of the activities where they should be used.<br>- Generate a list of sensory perceptions and associate them with each of the activities where they should be presented.<br>- Generate a list of interaction styles for the experience and relate them to each of the activities where it should be available to the user.<br>- Generate a list of gamification elements and relate them to each of the activities. It is advisable to consider elements such as [29]: goals and objectives, rules, narrative, freedom to choose, freedom to make mistakes, rewards, feedback, visible status, cooperation and competition, time constraint, progress, and surprise.<br>- Generate a list of the emotions you want to evoke in the user in each of the activities.<br>- Represent the learning objectives as zones and the activities as nodes that pertain to each of the objectives. Connect the activities with a line containing a time arrow that denotes the sequential nature of activities and represents each activity: digital media, sensory perceptions, interaction styles to be displayed in each activity, emotions, and gamification elements.<br>- Support each of these activities with the information that the team considers necessary and relevant to describe it; these include the following: materials to evaluate the learning objectives, outputs or deliverables of the activity, and success criteria of the activity, among other useful information that contributes to confirming that the students acquired a certain competency. All information produced as a result of the application of other techniques should be considered within the activity information.<br>- Analyze the map and its results. Look for ideas related to the experience and the activities, the display of its digital media, perceptions and the possibilities of interaction that it can offer.<br>- Summarize the findings and share them with the other team members, discussing whether there are opportunities to make the user journey more engaging and enjoyable during the experience.<br>- Keep the Journey Map produced. | Office and design tools for the creation of the Journey Map design. |

**Table 2.** *Cont.*

| Activity | Code Technique | Technique | Steps for Its Application | Tools |
|---|---|---|---|---|
| A3 | T7 | Matrix for preliminary definition of digital media type | - On the horizontal axis, define the different types of digital media, and on the vertical axis, relate the points defined in the history of the experience where multimedia content will be displayed.<br>- In the intersection points of the matrix, define favorable and unfavorable concepts for the use of each digital media in each of the milestones defined in the history. For each digital media and each milestone, design a weight from 1 to 100.<br>- Selection of digital media for the potential production of multimedia content. | - Office tools to support the production of the matrix.<br>- Materials and resources for the recreation of the matrix in a specific physical space. |
| A4, A6 | T8 | Analysis of information classification. | - Define the set of test participants and information categories for the multimedia experience.<br>- Allow the participants to make groupings for information categories into the sets they consider necessary for the open approach.<br>- Define the grouping sets for the categories in the closed approach.<br>- Perform qualitative and quantitative analysis of the results as required. | - Information sorting cards. |
| A4 | T9 | Design for content structure and information. | - Create a list of information components for the multimedia experience and classify them with labels by title, author, images, audio, video, and documents, among others.<br>- Design the structure and schemas that represent the flow of information throughout the experience.<br>- Define the organization, labeling, navigation, and search systems that should be offered throughout the experience.<br>- Establish the potential points of interaction in the content with the user.<br>- Socialize this structure and schemes with the other architects and make adjustments. | - Office tools to support the design of the content and information structure. |
| A4, A5 | T10 | Wireframes | - Build representations of potential user interfaces, considering the experience.<br>- Define the elements that should be interactive with the user through the sketch of the interfaces.<br>- Associate digital content for each wireframe component.<br>- Link potential sensory perceptions and interaction styles, and attach them to each wireframe element.<br>- Socialize the proposal and adjust.<br>- Retain low-fidelity prototypes of the wireframes produced. | Authoring and office IT tools to support the production of the listing. |

**Table 2.** *Cont.*

| Activity | Code Technique | Technique | Steps for Its Application | Tools |
|---|---|---|---|---|
| A5 | T11 | Asymmetric clustering matrix between sensory perceptions, interaction styles and digital media. | - Define the sensory perceptions and interaction styles to be analyzed against the digital media selected to produce multimedia content.<br>- Define the relationship between the sensory perception and interaction digital media most suitable for the multimedia experience.<br>- Create the asymmetric matrix with the three dimensions described.<br>- Define a score for each relationship, ranging from 1 to 3. Here, 1 represents a poorly adequate relationship, 2 represents a moderately adequate relationship and 3 represents a very adequate relationship.<br>- Construct the asymmetric matrix with the parameters and assign scores for each of the relationships.<br>- Capture ideas and share them in open discussions with the work team. | Office IT to support the production of the matrix. |
| A6 | T12 | Behavioral prototyping | - Identify the specific Journey Map activity you wish to simulate and plan the simulation of the situation, including the necessary elements, actors, and objects involved in the activity.<br>- Prepare the environment for the simulation, by finding or producing a physical or virtual environment, and state the key behaviors you want to understand (physical, cognitive, social, cultural or emotional). Use props to represent the concepts and support participants to interact with them.<br>- Invite users to participate in the simulation of the specific activity designed, defining whether they should perform it individually or in a group, and guide them during the process.<br>- Observe, document, and consult with the users regarding their behaviors and how they affect the physical, cognitive, social, cultural, and emotional aspects of the user.<br>- Analyze user behavior patterns from videos and notes taken, and review what concepts can be improved in the experience design due to the application of the technique; then, iterate against any adjustments the team feels should be made.<br>- Retain the low-fidelity prototypes used to capture user behavior patterns. | - Office IT tools to take notes during the process.<br>- Resources for audiovisual capture of the simulation process, carried out by users with the prototype. |
| A6 | T13 | Concept prototypes | - Identify the concept or idea related to the experience you wish to test and evaluate whether it is susceptible to prototype and what its usefulness would be.<br>- Define a space where the prototype construction and testing can take place.<br>- Review the prototype that is associated with the experience, test it with your team and with a group of users and discuss it, based on the principles of the initial experience design, user needs, interests and expectations.<br>- Adjust the prototype or replace it with a new one, feeding it with the feedback obtained.<br>- Summarize the analysis of how the prototype evolved from an original idea to the desired state, socializing these results with your team, customers and sponsors.<br>- Keep the prototypes. | - Materials and tools that allow the creation of the prototype. |

2.4.2. Responsible Route

The Responsible Route provides the work team with an analysis process that allows them to identify critical factors related to the Value Sensitive Design [30]. The objectives of this path are as follows:

- To reference the main laws and regulations associated with the educational context at a local, regional, national, and global scale, and which may influence the design of the IME.
- To recognize the factors related to indirect and long-term effects that may occur because of the use of IME.
- To recognize and document possible patterns that respond to the design of the multimedia system development context and other existing solutions, as well as behavioral patterns in people.

The Responsible Design Route is aimed at identifying local, regional, national and global regulations and policies that can potentially influence the IME and must be taken into account to ensure a solution aligned with them. Likewise, it recognizes the impacts that the solution may produce, in the medium and long term, as a result of its prolonged use in other contexts, such as in organizations or communities; it helps to identify patterns in these contexts, in order to favor the design of the IME [2]. The active and collaborative participation of the following stakeholders is essential for the implementation of the Responsible Route:

- Rector, academic coordinator or pedagogical advisor: provides information on regulations, basic learning rights, learning standards in different areas (science, social, etc.) and current laws on education issued by government institutions, which the work team must consider in the design of the multimedia experience.
- ICT manager: accompanies the work team for the correct inclusion of the regulations in the elements that make up the IME, such as digital content, learning activities, game mechanics, gamification, and others.
- Work team: in charge of considering the current regulations in the pre-production of the multimedia experience.

Table 3 presents the set of activities involved in the Responsible Route applied to carry it out. In addition, Table 4 presents a series of steps associated with the techniques and tools used by the team. The basis of these techniques and the steps for their application are based on approaches and frameworks related to tools for the ethnography of the network [31], techniques based on Design Thinking [26], VSD (Value Sentitive Design) practices [32] and architectural models that help to identify patterns [25].

**Table 3.** Activities of the Responsible Route.

| Activity Code | Activities |
|---|---|
| A7 | Identification, analysis and classification of the laws and regulations in force at the global, national, and regional levels that may influence the design of the solution. |
| A8 | Identification of factors related to indirect and long-term effects that may occur as a result of the use of the multimedia system. |
| A9 | Analysis, identification, and classification of possible patterns that respond to the conception of the development context of the multimedia system and other existing solutions, as well as behavioral patterns in people. |

**Table 4.** Detail of the techniques associated with the Responsible Route in elementary education.

| Activity | Code Technique | Technique | Application Steps | Tools |
|---|---|---|---|---|
| A7 | T14 | Identification of policies and regulations | - Conduct a brainstorming meeting with subject matter experts that will help identify the following:<br>• Current policies and regulations associated with the educational context and organization(s) involved, represented by sponsor(s) and client(s).<br>• State policies that must be considered and potentially constrain the design of the solution.<br>• Basic learning rights.<br>• Learning standards in different areas (science, social studies, foreign languages, etc.).<br>   * Test results applied by the government.<br>   * Recommendations to avoid violent, sexist, or discriminatory content.<br>• Policies and regulations that potentially affect the design of the experience, given their influence on the user.<br>- Discuss and analyze the results obtained with the work team. | Brainstorming meeting. |
| A8 | T15 | Non-directed use | - Identify possible uses for the multimedia system other than those originally conceived by its designers.<br>- Analyze what kind of stakeholders may come to make a different use of the system and estimate their possible motives.<br>- Identify whether these motives may influence ethical, moral, and security issues that may affect users.<br>- Meet and discuss the results of the analysis with your team. | Use the envisioning cards kit [33]. |
| A9 | T16 | User feedback analysis | - With a spreadsheet, compile the information obtained as a result of the questionnaires, surveys and interviews conducted with the users.<br>- Reduce and organize the data related to the research, through information topics that the team wishes to analyze (groups, segment of groups or individual responses), and choose topics for their respective comparison.<br>- Define the type of search you wish to perform, establishing a set of keywords that represent the topic of interest to be investigated.<br>- Generate a visual code that facilitates quick understanding of the search results, e.g., user responses can be color-coded for age, gender, etc.<br>- Analyze the coded information and identify patterns and insights resulting from the study of the data obtained.<br>- Summarize the results obtained and discuss them with the team. | Matrix of user feedback patterns. |
| A9 | T17 | Pattern identification | - Make sure you understand the context in which the deployment of the multimedia experience will take place.<br>- Study the scenario and identify patterns at the abstraction level of the context.<br>- Identify 'scenario' patterns that establish the context. To do this, approach the problem at its highest level of abstraction by identifying patterns that meet your needs.<br>- Work within the context to identify patterns at a lower level of abstraction that will further assist in the design of the solution. To do this, try to subdivide the problem into more concrete problems and identify patterns at that level.<br>- Research and review documentation of projects that offer solutions to similar problems and contrast the patterns identified with those found in the review.<br>- Establish a baseline of preliminary patterns and discuss your findings with other team members. | - Pattern matrix. |

*2.5. Relationship between Practice Techniques and Gamification Elements*

In an educational context, gamification refers to the use of game elements to engage students, motivate them to action, and promote learning and problem solving [34]. In that sense, the application of the practice for designing IME in elementary education should lead to the design of experiences that increase students' motivation and performance. Therefore, the techniques used in the design of the multimedia experience are an important basis for implementing gamification elements. It is not necessary to consider all the techniques provided by the practice [24], but it is necessary to take those that, by their characteristics, may be more valuable in implementing the gamification elements in the sought-after learning experience. Based on this, Table 5 presents the relationship between the techniques used in the practice and their contribution to the gamification elements.

**Table 5.** Contribution of the techniques to the gamification elements.

| Code Technique | Technique | Contribution to the Elements of Gamification | Related Hypothesis |
|---|---|---|---|
| T1 | Video to support ethnography | Detection of patterns in the context and behavior of users when performing individual or group activities, which can be used to define gamification elements that are familiar to them. | H1, H2 |
| T2 | Definition of users | This technique represents a fictitious user of the experience, its demographics, interests, motivations, needs, frustrations, and other relevant information, which can be analyzed to define gamification elements in the learning activities. | H2 |
| T3 | Metaphor and analogy production | Definition of the best metaphors and analogies for the design of the story/narrative of the multimedia experience that will be interesting and surprise users. | H2 |
| T4 | Storyboard generation | Design of the structure and graphic representation of a story, characters, scenarios, and flow of events of the interactive multimedia experience, to direct the student's progress, accompanied by the gamification elements defined in the learning activities. | H1 |
| T5 | Map of compelling experiences | Creation of a novel experience that generates student motivation through the definition of challenges or problematic situations to be solved. | H1 |
| T6 | Journey Map design for the experience | Description of key moments (milestones) of the multimedia experience. Each milestone defines the digital media, sensory perceptions, modes of interaction, emotions expected to be evoked in the user, and gamification elements. This technique helps to confirm clarity in the purpose of the activity and the motivation generated by the gamification elements, to confirm that the learner's effort is well directed. | H1, H2 |
| T7 | Matrix for preliminary definition of digital media type | Definition of digital media (such as text, audio, images, animations, videos, 3D objects, and others) to support the story of the multimedia experience and provide feedback to the user's actions. | H2 |
| T8 | Analysis and classification of information. | Grouping of gamification elements according to the competencies expected to be achieved in the learning activities defined in the multimedia experience. | H1 |
| T9 | Design for content structure and information. | Organization of gamification elements according to the flow of actions, digital content and other information offered by the multimedia experience. | H2 |
| T10 | Wireframes | Representation of the user interfaces in the experience through low-fidelity prototypes, which allow us to understand the flow of actions in the learning activities, represent and detail the defined gamification elements, and, among other design aspects, to ensure the achievement of the learning objectives and an attractive design. | H1 |
| T11 | Asymmetric clustering matrix among sensory perceptions, interaction modalities, and digital media. | Definition of the sensory perceptions to be stimulated by the user throughout the experience, and the interaction styles to be used in front of the digital media and multimedia content, which contribute to the achievement of the activities and learning objectives. | H1 |
| T12 | Behavioral prototyping | Observation and documentation of user behavior with respect to the interaction and use of the gamification elements; this is used to determine if their attitude is favorable to the design of the interactive multimedia experience. | H2 |

**Table 5.** *Cont.*

| Code Technique | Technique | Contribution to the Elements of Gamification | Related Hypothesis |
|---|---|---|---|
| T13 | Concept prototypes | Validation of the design of the multimedia experience to obtain feedback from stakeholders on the satisfaction of needs and expectations. It makes the gamification elements and their contribution to the learning objectives visible. | H1 |
| T14 | Identification of policies and regulations | Identification of policies, laws, basic learning rights, audiovisual content suitable for the target audience and other regulatory norms by the state that should be considered in the design of the multimedia experience in elementary education. | H1 |
| T15 | Non-directed use | Identification of other uses for the multimedia experience and gamification elements, other than those originally conceived by the work team. | H2 |
| T16 | Analysis of user feedback | Analysis of user data and identification of patterns that contribute to the definition of game elements, which can be more valuable to the interactive multimedia experience and generate student engagement with the activity they participate in. | H1 |
| T17 | Pattern identification | Analysis, interpretation, and detection of the behavioral patterns of potential users that can be taken as a basis for the definition of gamification elements in the multimedia experience; this is in order to achieve a positive experience and achieve learning. | H1, H2 |

*2.6. Case Study*

In Colombia, the largest repository of digital resources to support elementary and secondary education can be found on the 'Colombia Aprende' website [35]. This national resource is mainly used by public schools as a tool for general-purpose learning experiences. Specifically, for English language learning, the repository offers a limited number of resources and features tools that have been produced under general purpose guidelines, established by the Colombian Ministry of National Education [36]. In other cases, existing solutions are available on the Internet, and it is assumed that they can be useful in supporting different academic teaching/learning activities. Some of these resources could potentially be used for the design of gamification strategies. Meanwhile, others are more oriented towards serious games, as is the case with "B (the) 1: Challenge" [37].

As a result of applying the practice for the design of IME in elementary education, the experience called 'Coco-Shapes' was developed for transition students of the La Fontaine School in Cali, Colombia. Topics that were considered in the transition grade included objects, descriptions, qualitative and quantitative terms, connectors, and some verbs. As part of the school's educational strategy, Coco-Shapes contributes to the specific topics of colors, shapes and counting. Coco-Shapes consider a set of objectives aligned with the Basic Learning Rights, as well as the needs of the school's teachers. The learning objectives were defined as follows:

1.  To develop interactive listening and English interpretation skills in transition grade students at La Fontaine School.
2.  To identify primary and secondary colors and their pronunciation in the English language.
3.  To identify a set of five geometric figures, their shape, and pronunciation in the English language.
4.  To develop mathematical notions associated with counting numbers from 1 to 20 for transition grade students, their notation, spelling, the quantities they indicate, and pronunciation in English.

After carrying out an inquiry process with potential stakeholders (parents, schoolteachers, and pedagogues), the following needs were identified:

- School administrators need interactive resources that incorporate activities of interest that promote the learning of English and associated content (vocabulary related to geometric figures, colors and counting).

- Managers need the proposed system to make use of electronic equipment, such as tablets, which are available in the classroom and/or playful objects that promote learning.
- Managers need the system to be designed to operate in the available space, expected classroom lighting conditions, and number of students in the classroom.
- Students need follow-ups in the process of learning English.
- Teachers need to have access to the record of activities performed with the system, in order to make pedagogical decisions based on the records.
- Teachers need these interactive resources to reinforce the topics studied in class, specifically colors (primary, secondary, black, and white), shapes (squares, circles, rectangles, triangles, and stars) and counting from 1 to 20, for transition grade students.
- Teachers need to match the activities to each student's learning level, to achieve effective English learning in an interactive way.

In this sense, hypotheses H1 and H2, which guide this research, are transversal to the learning objectives and the needs detected. Hypothesis H1 allows the investigation of the fulfillment of the learning objectives using the multimedia system Coco-Shapes, which includes a series of gamification elements; these support the teaching/learning of the English language, specifically themes associated with colors, shapes and counting for transition grade students. On the other hand, hypothesis H2 allows the discussion and investigation of the contribution of gamification elements in a multimedia system for transition grade students, so that they become familiar with the use of technologies in their English language learning process. The latter allows the teacher to plan activities and learning strategies for the classroom [38].

Based on the needs of the stakeholders, the work team defined a series of learning activities, according to the students' previous knowledge and experience in the English language. These activities had three levels of difficulty, adjusted to the student's progress, and were associated with the themes of colors, shapes and counting. The multimedia experience 'Coco-Shapes' would allow students to learn and/or practice the English language through physical objects (buttons and shapes) and digital resources, deployed on a tablet. Regarding the activities with geometric shapes, children had to insert the corresponding shapes into a box. In the color activities, students had to press the colored buttons that were also found on the physical object. Finally, in the counting activities, students had to select the correct option on the screen.

On the other hand, the students' results can be consulted by the teachers of the course, in order to evaluate the progress of each one of them. Finally, the system was adapted to the conditions of the context of use, such as lighting or desk size, in addition to using the available resources, such as tablets, computer rooms, and the internet.

*2.7. Work Products*

A relevant work product in the application of the Creative Route is the Journey Map (resulting from the T6 technique), which details the design of the IME. It allows the definition of the sensory perceptions expected to be stimulated in the users, the digital media that support the story, the interaction styles, and other relevant information, in a series of milestones (or important moments) in the experience. Based on this technique, the stakeholders defined the gamification elements of Coco-Shapes, which were as follows: goals related to pedagogical needs in the classroom, freedom to choose between three learning topics, feedback to user actions, as well as freedom to make mistakes.

According to the scope of the Coco-Shapes solution, we were given the structure and narrative flow of events in the story to be developed from the problem and learning objectives; these described the scenarios and main character, where the user finds digital media content, the stimulating sensory perceptions and where to find the options for interaction with the experience. As a result, there is a Journey Map made up of 12 milestones. Due to the length restriction on this article, we have detailed three milestones from the Journey Map below.

**Milestone 4: Development of figure activity**

Description: Coco is walking through the park and must play hopscotch (see Figure 2). To pass the hopscotch task, the student must insert the geometric figures shown in each of the boxes in the physical object, in the defined order. The objective is to strengthen the student's ability to identify the shape of the figures.

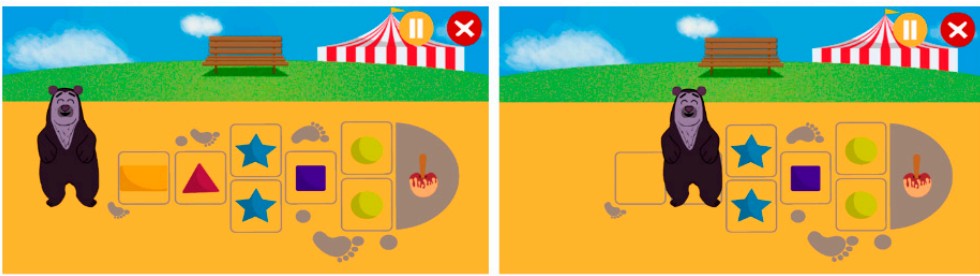

**Figure 2.** Level 1 figures for transition grade students.

Digital media:

- Graphics: There are 2D animations and illustrations, in which the character of the experience performs the necessary actions during each activity and shows the context where they take place.
- Text: English statements necessary to carrying out the figure activities proposed in the interactive multimedia experience.
- Audio: Sound effects that accompany the actions of the main character and a voice-over that provides feedback to the students according to their answers.

Sensory perceptions: The student who performs the activity observes the actions performed by the main character during its development, paying attention to the sound effects provided by the system and the voice-over; this provides feedback to the answers given by the student, through the physical objects of figures integrated in the interactive multimedia experience. Visual, auditory, and tactile sensory perceptions are involved.

Modes of interaction: Interaction with the physical objects involved in the interactive multimedia experience required the student to insert the figure that they considered to be correct during the development of the activities.

Emotions: Students' joy when performing the activities proposed by the interactive multimedia experience.

Regulations considered in the design:

- Basic English learning rights [39], associated with recognizing simple instructions related to their environment, and associating pictures with word sounds related to their home and classroom.
- Childhood and Adolescence Code, Law 1086 of 2006, Article 33 [40]. The right to privacy is vital to ensure that student data stored in the experience (such as name, date of birth, and grade level) does not violate their privacy in any way.
- Resolution No. 3158 of 2007 [41] establishes the safe and normative design of physical objects that integrate the multimedia experience. In the design and production of the physical objectives, the use of non-flammable materials, electrical voltage (not exceeding 24 volts), no sharp edges of glass or metal, and no access to electronic components (such as batteries) was considered by the students.

**Milestone 7: Development of exploratory level of colors**

Description: Coco is preparing cotton candy in secondary colors, for which the student must press the button of the two primary colors requested, through audio and text in English, in the requested order (see Figure 3). The objective is that the student develops the ability to read and listen to the secondary colors in English.

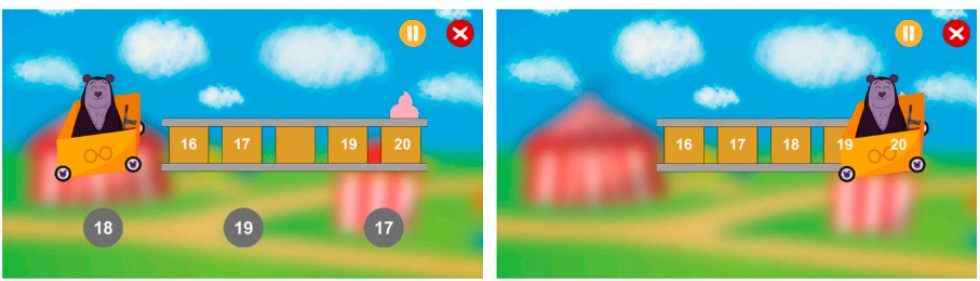

**Figure 3.** Exploratory level of colors for transition grade.

Digital media:

- Graphics: There are 2D animations and illustrations, in which Coco makes the combination of primary colors to obtain secondary colors, and the context where the activity takes place is shown.
- Text: Instructions during the development of the activity.
- Audio: Sound effects for the context in which the activities and actions performed by the character take place, voice-over requesting primary colors, and providing feedback on the actions performed by the students.

Sensory perceptions: The student who performs the activity observes the actions that the main character of the interactive multimedia experience performs during the development of the proposed activities, listens to the sound effects of the actions performed by the character and the context in which the activity takes place, and responds to the activities of the experience through colorful physical objects.

Interaction modalities: Interaction with the colored physical objects that integrate the interactive multimedia experience, the student presses the colored buttons and the character of the experience requests them.

Emotions: Students' monitoring of the activity, proposed by the interactive multimedia experience, to understand the topics presented during its development.

Regulations considered in the design:

- Decree 975 of 2014, Article 3°, Rights of children and adolescents regarding information and advertising [42]. This regulation is relevant for the development of multimedia content that will be part of the experience, ensuring that it does not represent any form of violence, harassment, or discrimination.

**Milestone 11: Development of counting activities**

Description: Coco must reach the cotton candy at the end of the road. Then, he must complete the number sequence shown in the roller coaster boxes (see Figure 4). The student must count the numbers and identify the missing number, to select the correct option from those available on the screen. The objective is to develop mathematical notions associated with counting, with numbers from 1 to 20, their notation, spelling, the quantities they indicate, and pronunciation in English.

**Figure 4.** Level 1 Counting Activities for Transition Grade Students.

Digital Media:

- Graphics: Illustrations and 2D animations in which the main character in the experience is seen performing different actions proposed during the development of the counting activities; the answering options are shown on the screen.

- Audio: Sound effects related to the context in which the experience takes place.
- Text: Activity statements and answering options for counting activities.

Sensory perceptions: The student performing the activity observes the actions performed by the main character of the experience and triggers the questions for the activity, listens to the sound effects of the context of the activity, and selects the answer by pressing the corresponding button on the screen.

Modes of interaction: Direct interaction with the tablet's screen to select the button with the answering option the student chooses.

Emotions: Students' joy when performing the activity proposed by the interactive multimedia experience.

Regulations considered in the design:

- Basic mathematics learning rights [43]. They are important for the design of counting activities proposed in the multimedia experience, with the objective that students recognize the use of numbers to perform addition and subtraction operations, while being able to compare numbers and associate these with quantities.

As a complement to the conception of the interactive multimedia experience, the Responsible Route allowed (i) the establishment of a series of regulations that could influence the development of the solution, as described in each milestone; and (ii) the identification of the uses that the multimedia experience will have, which do not correspond to the purpose for which it was designed. These uses are the implementation of the experience in support workshops by governmental or citizen entities, as well as the mass marketing of the multimedia experience by private entities with economic interests and for market recognition.

Finally, the path allowed this study (iii) to identify the patterns in the context in which the experience is developed, as well as the behavioral patterns identified in the users at the time of designing the interactive multimedia experience. The behavioral patterns highlighted are as follows: (a) using tangible elements in the system generates elevated expectations in the users, (b) providing clear instructions in the activities contributes to their effective completion, (c) there is a high occurrence of errors in topics related to the combination of colors, addition, and subtraction, and (d) users respond quickly to activities involving audio instructions, among others.

### 2.8. Evaluation Method

To test the above hypotheses, the evaluation process was carried out through a set of activities divided into three stages, as presented below. Stages 1 and 2 were carried out by two different teachers, who have the same level of proficiency and academic qualifications for teaching English; they are also the usual teachers of the children.

Stage 1. Control group without use of Coco-Shapes

- The teacher teaches 18 transition students the topics of colors, figures and counting in the morning (07:00 to 12:00), following the methodology defined by their experience and the school's approach.
- Evaluation of the appropriation of knowledge through a mechanism defined by the teacher.
- The teacher stores the data for later analysis.

Stage 2. Experimental group using Coco-Shapes

- In the afternoon (12:30 to 17:30), the teacher teaches 18 transition students the themes of colors, figures and counting, following the methodology; this is defined by his experience and using Coco-Shapes to accompany the activities associated with each theme.
- The application provides the teacher with a module in which they can consult quantitative information about each student; this includes the interaction time with the system, the number of levels achieved, the number of successful activities, the number of activities in which they made mistakes, the time spent on each activity, and the level.
- The teacher performs a field observation and takes note of the expressions and comments of the students.
- Evaluation of knowledge appropriation through the same mechanism defined in Stage 1.

- The teacher stores the data for later analysis.

Stage 3: Discussion

- From the results obtained by the students in the control and experimental groups, the teacher contrasts the results between the first assessment (stage 1) and the second (stage 2).
- A discussion takes place with the teachers to learn how they perceived the contribution of the multimedia experience to the students' English language learning process.

### 2.9. Population and Sample

For the application of the evaluation methods described above, the model suggested by Sauro and Lewis [44], for the calculation of the sample size, was considered:

$$n = \frac{\frac{z^2 \times p\ (1-p)}{e^2}}{1 \pm \frac{z^2 \times p\ (1-p)}{e^2 N}} \tag{1}$$

where $n$ is the sample size, $N$ is the population size, $e$ is the margin of error (which is a percentage expressed in decimals), $z$ is a statistical parameter that depends on the confidence level, $p$ is the probability that the studied event occurs, and $(1 - p)$ is the probability that the studied event does not occur. In this case study, the population size is 41 people, made up of two groups of students: there are 18 students from the morning session and 18 from the afternoon session. Likewise, the process carried out with the students of both shifts is guided by a total of 5 teachers, with at least 3 years of teaching experience. Given that the desired confidence level is 95% with a margin of error of 5%, the minimum sample size should be 38 people. However, in the experimentation process, all 41 people participated.

The students are people who belong to the same commune that the school is located in, so they occupy the same socioeconomic stratum and possess similar social and cultural conditions. The students are in an age range between 4 and 5 years old, with similar knowledge of and experience with the use of Information and Communication Technologies.

## 3. Results

The evaluation process allowed the collection of a series of data on the interaction of the children with the multimedia experience, in order to complete the levels of the learning activities. The afternoon group of 18 students acted as the experimental group, where they had the opportunity to work on each of the three levels with the multimedia experience offered by Coco-Shapes. Meanwhile, the morning group acted as a control group, where the topics associated with each of the three levels were approached in the conventional way by the teacher, using traditional resources and media in the classroom, without the use of Coco-Shapes.

The results obtained in both the experimental and control groups are based on the five questions asked by the teachers to each of the students. For the students in the experimental group, these questions were applied after the use of Coco-Shapes. In the case of the students in the control group, the same five questions were applied following the conventional activities of the teacher in the classroom; this included the use of commonly-used traditional media and instruments, such as songs, repetitive activities, illustrations, and videos.

Table 6 shows the results obtained. The numbers of students are distributed according to the results obtained for the experimental and control groups, based on the number of correct answers obtained, from 5 to 0. In addition, the range in which there is a higher concentration of correct answers is presented for further analysis.

Figure 5 shows the comparison between the results obtained from the experimental and control groups, for each of the three topics in level 1. A trend can be observed in

the experimental group, which shows that students present a greater number of correct answers with the use of Coco-Shapes, in contrast with the results of the control group.

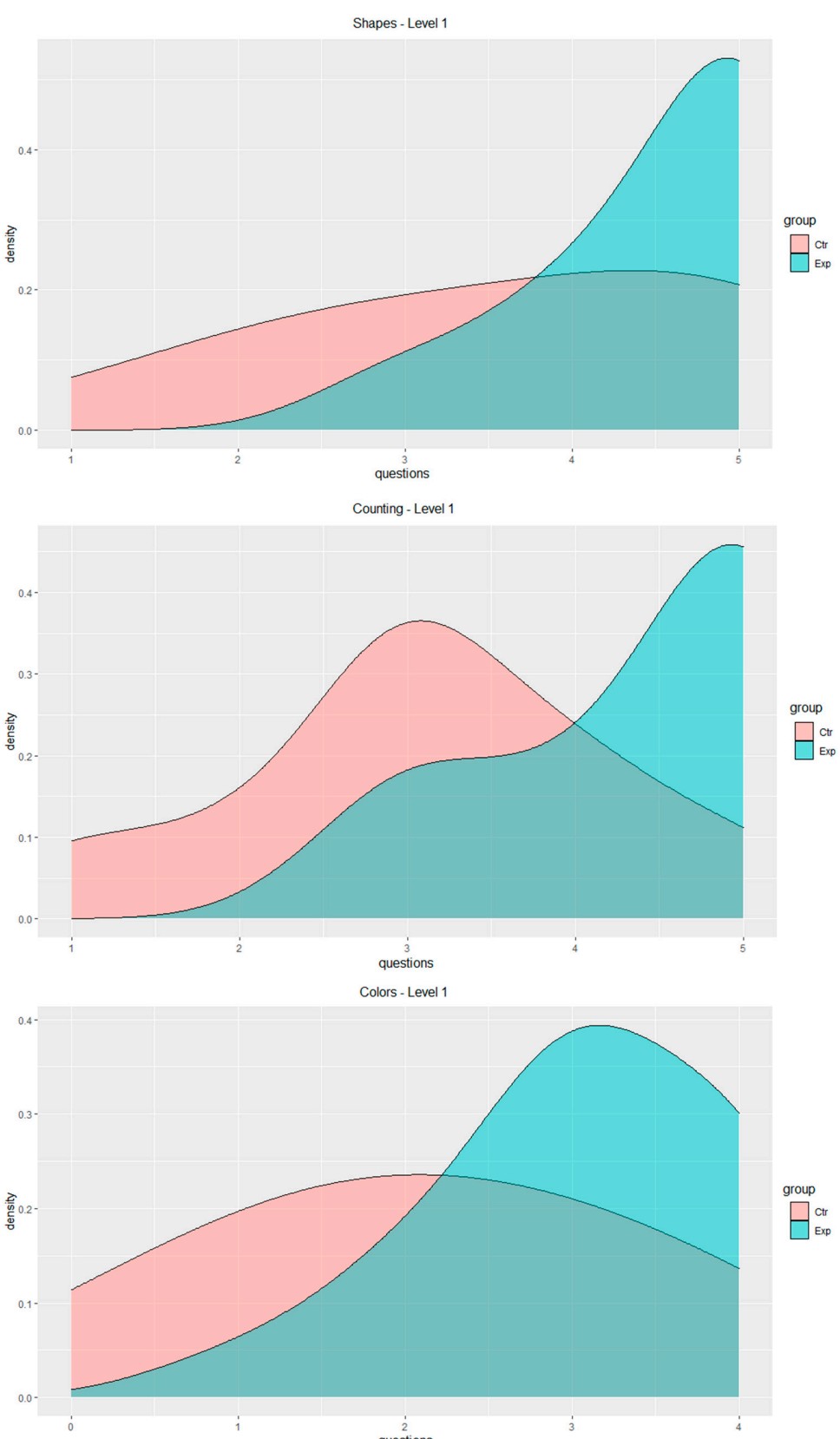

**Figure 5.** Comparison of results in the three themes of level 1.

A similar trend in results is observed for level 2, which can be seen in Figure 6, where the experimental group shows a higher degree of correctly answered questions because of the learning assessment using Coco-Shapes.

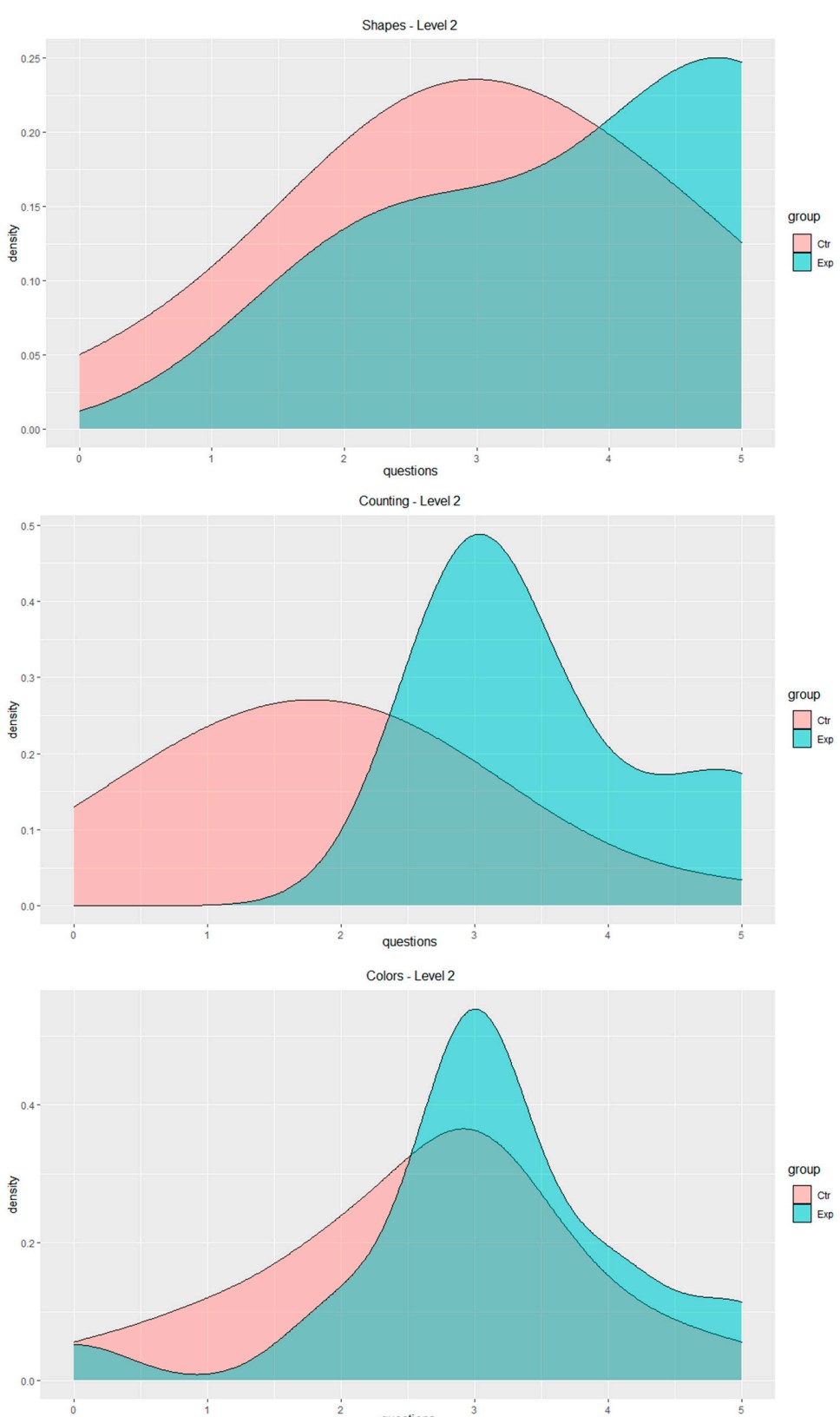

**Figure 6.** Comparison of results in the three themes of level 2.

In level 3, a trend similar to that in levels 1 and 2 is also observed; therefore,, in this case, the results of the control group, for the topics of figures and colors, show a better trend than in the previous levels, as shown in Figure 7.

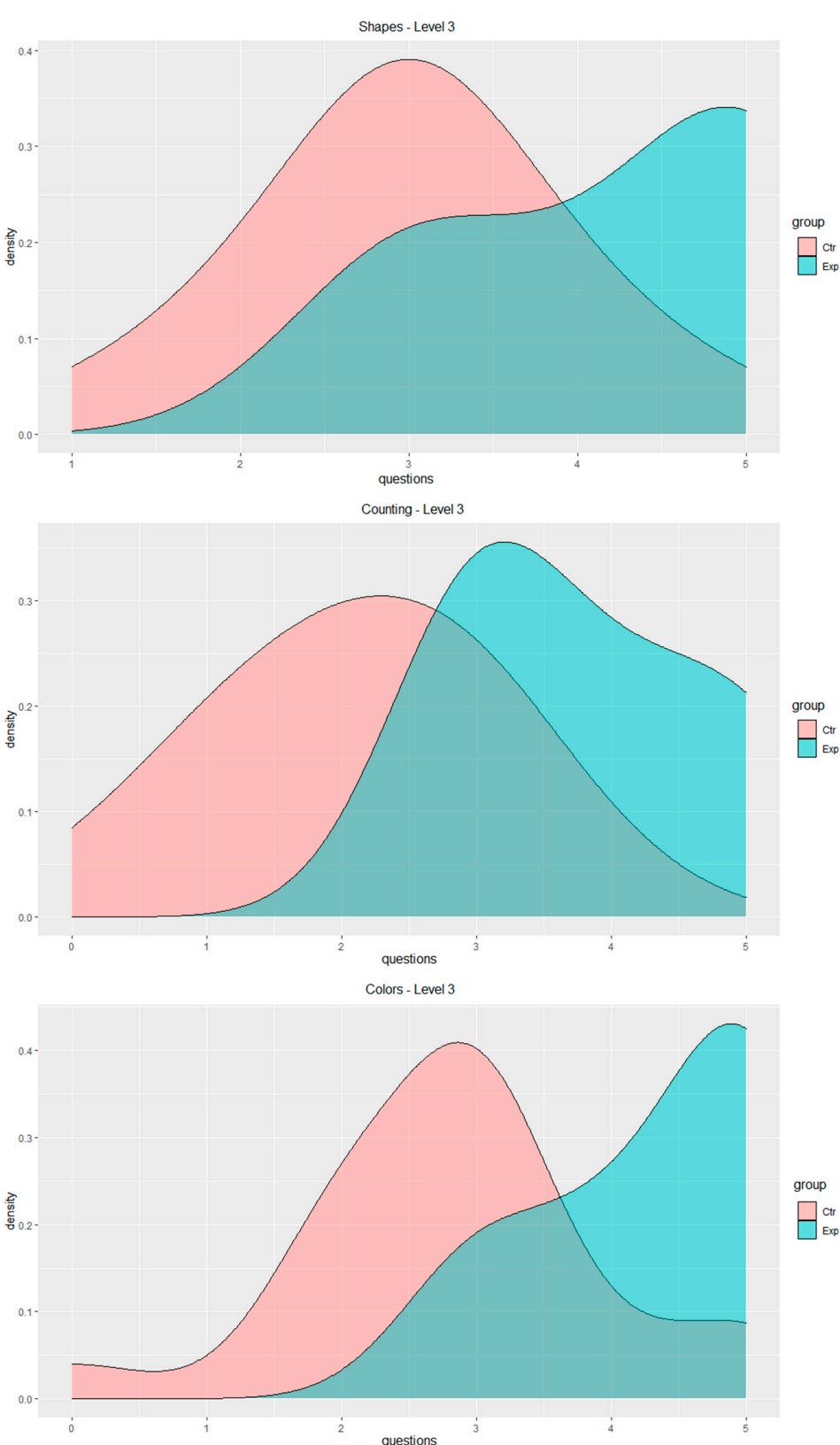

**Figure 7.** Comparison of results in the three themes of level 3.

**Table 6.** Experimentation results.

| Level | Thematic | Number of Students | | | | | | | | | | | | | |
| | | Experimentation | | | | | | Concentration Range | Control | | | | | | Concentration Range |
| | | 5 | 4 | 3 | 2 | 1 | 0 | | 5 | 4 | 3 | 2 | 1 | 0 | |
| Level 1 | Figures | 12 | 4 | 2 | 0 | 0 | 0 | 4–5 | 7 | 3 | 4 | 3 | 1 | 0 | 3–5 |
| | Counting | 11 | 3 | 4 | 0 | 0 | 1 | 4–5 | 2 | 4 | 8 | 2 | 2 | 0 | 3–4 |
| | Colors | 0 | 6 | 8 | 3 | 1 | 0 | 3–4 | 0 | 3 | 4 | 5 | 4 | 2 | 1–3 |
| Level 2 | Figures | 10 | 1 | 3 | 4 | 0 | 0 | 3–5 | 3 | 3 | 6 | 4 | 1 | 1 | 2–4 |
| | Counting | 4 | 2 | 12 | 0 | 0 | 0 | 3–4 | 1 | 0 | 4 | 6 | 5 | 2 | 1–3 |
| | Colors | 2 | 3 | 10 | 2 | 0 | 1 | 3–4 | 1 | 2 | 8 | 4 | 2 | 1 | 2–3 |
| Level 3 | Figures | 10 | 2 | 6 | 0 | 0 | 0 | 3–5 | 1 | 3 | 10 | 3 | 1 | 0 | 3 |
| | Counting | 5 | 4 | 9 | 0 | 0 | 0 | 3–5 | 0 | 1 | 6 | 6 | 4 | 1 | 1–3 |
| | Colors | 10 | 4 | 4 | 0 | 0 | 0 | 4–5 | 2 | 1 | 9 | 5 | 0 | 1 | 2–3 |

## 4. Discussion

In this study, the researchers sought to understand how an IME based on gamification elements contributes to increasing user learning (H1), and how it contributes to achieving a favorable attitude towards the technology; this significantly predicts the user's reactions (H2). The results of the experimentation, in particular the concentration ranges in the results of the evaluation exercise for the teachers with the experimental and control groups, provide optimistic evidence that an IME, based on gamification elements, contributes to the knowledge that can be acquired through learning activities conceived by a work team.

Regarding the first hypothesis (H1), Table 6 and Figures 5–7 allow us to compare the results obtained by the experimental and control groups, where the former generally achieved better concentration ranges, in response to a high valuation for the teacher. This suggests that the use of Coco-Shapes, as an IME conceived through a practice for designing IME in elementary education, includes gamification elements that create value for the user. Therefore, it contributes to the learning process. The work paths, activities, and techniques of the method allow this study to define the gamification elements that correspond to the learning objectives and their incorporation of attractive and challenging learning activities; this is in order to guide the student's experience towards the development of the expected competencies.

As part of the practice proposed in this study, the T6 technique suggests the definition of concrete gamification elements that contribute to the conception of a playful and fun multimedia experience, so that students can enhance their skills. However, during the evaluation process using Coco-Shapes, it was observed that not all children understood the instructions given through the audio, due to the limited vocabulary of the transition grade; however, the digital contents (such as the introductory animation) contributed to the understanding of the objective and rules of the activity. Therefore, the T6 technique gives a work team the opportunity to analyze the relationship between components of the IME with the gamification elements that accompany it.

In the case of the counting theme, although children in the transition grade do not know how to subtract, the instructions of the activity and the roller coaster scenario are of interest for the child to practice and develop the first notions of counting. The children take on the activity as a challenge, looking for ways (counting with their fingers and in English) to achieve good results. In addition, the visual and auditory feedback provided at each attempt allows the child to understand the dynamics of the activity, so that they stop guessing the answers and the children gain knowledge of the subject. This is reflected in the evolution of the results for which the teacher was consulted, in regard to the use of Coco-Shapes.

Considering the second hypothesis (H2), the evaluation process suggests that students' attitude towards the use of an interactive multimedia experience is favorable, to the extent that it matches their interests, needs, and expectations, as intended by the proposed practice

(see Section 2.4). In the process of understanding the student's attitude towards technology and learning outcomes, the findings during the evaluation reveal a positive correlation; children are excited to know that the next level of an activity is more difficult than the previous one. Similarly, children seem excited to see that the narrative of the multimedia experience is based on an amusement park, has a character that is familiar to them, and includes a roller coaster and fairground rides, among other elements that generate surprise for them. This is a product of the Creative and Responsible work paths proposed in this study, which contribute to the conception of a solution aligned with the users' needs.

The findings of this study confirm the two hypotheses proposed, providing a series of contributions for different stakeholders. First, the practice for the design of IME in elementary education provides the possibility for different stakeholders to direct their projects and initiatives to conceive, with a creative sense, solutions based on interactive multimedia experiences in a school context. Secondly, the techniques considered in this practice are related to the design of gamification strategies and well-defined elements, with which a positive impact can be achieved in IME and learning objectives can be pursued. Finally, the evaluation process allows us to observe that there is a statistically significant difference in the results' concentration ranges between the experimental group and the control group. This suggests that an interactive multimedia experience based on gamification, as is the case with Coco-Shapes, favors learning results and increased knowledge, in the context of elementary education, and influences the attitude and reactions of the students.

## 5. Conclusions

This study established a practice for the design of the IME in elementary education, under a value-sensitive design approach; this guides the work teams responsible for its realization via the clear definition of gamification elements in the solution, contributing to the creation of value for stakeholders. This practice seeks to align the needs of stakeholders, learning objectives, and constituent elements of the multimedia experience (such as the selection of digital media, ethical and moral factors to be considered, sensory perceptions, system interaction modalities, and interaction styles), as a basis to guide its subsequent production.

An interdisciplinary work team, guided through the Creative Route, will be able to carry out the process, accompanied by the techniques and tools that facilitate the conception of an IME and its alignment with the interests of elementary education institutions; the inclusion of gamification elements in the learning activities, with the purpose of influencing student behavior and increasing motivation, will also be performed by the work team.

The Responsible Route is aimed at identifying local, regional, national, and global regulations and policies that can potentially influence the IME, and must be considered to ensure a solution aligned with them. Likewise, it recognizes the impacts that the solution may produce in the medium and long term, due to its prolonged use in other contexts, such as organizations or communities. It also helps to identify patterns in a school context and in user behavior, which will favor the design of the IME in elementary education.

The practice for the design of the IME in elementary education has shown its effectiveness when applied as a strategy to include technology-mediated gamification in highly vulnerable communities. This offers an optimistic outlook for the development of further experiences that favor inclusive and quality education, providing learning opportunities to children belonging to low-income communities.

The proposed practice allowed a work team to conceive the IME called Coco-Shapes, which provided the teacher with a module to consult student data via their interaction with the system; this included interaction time, the number of levels achieved, the number of successful activities, the number of activities in which mistakes were made, the time spent in each activity, and the level. This information is useful for the teacher, since it allows them to monitor the learning process of the students, as well as define teaching strategies. However, for future work in the short term, it is important to consider aspects of learning analytics that contribute to the decisions that the teacher can make for the benefit of the

students, to a greater extent, as well as to analyze the contribution of gamification elements to the results achieved by the students, in greater detail.

The use of the practice demands considerable time for a work team because it requires the process of ideation and creation. Therefore, in future work, the definition of tools that apply artificial intelligence to assist a team in the execution of a series of techniques should be considered for review. Considering the above, there are software tools that can generate images and stories from a set of keywords, and these can be an input for the definition of scenarios, characters, and narratives, in the process of conceiving an IME.

**Author Contributions:** Methodology, C.A.P.; Investigation C.A.P. and A.S. All authors have read and agreed to the published version of the manuscript.

**Funding:** This research was funded by Universidad Autónoma de Occidente.

**Institutional Review Board Statement:** The study was conducted in accordance with the Declaration of Helsinki, and approved by the Institutional Review Board (or Ethics Committee) of La Fontaine School (approved on 9 December 2022) for studies involving humans.

**Informed Consent Statement:** Informed consent was obtained from all subjects involved in the study.

**Data Availability Statement:** Not applicable.

**Conflicts of Interest:** The authors declare no conflict of interest.

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
