# Peer review of "A Practice for the Design of Interactive Multimedia Experiences Based on Gamification: A Case Study in Elementary Education"

_sustainability, doi:10.3390/su15032385_

Round 1

Reviewer 1 Report

The paper presents a complex and long research, which aims to outline a design of interactive multimedia experiences (as mentioned by the authors).

The first thing that stands out is the fact that the title mentions "based on gamification", but the concept of gamification is not explored, keeping always vague information like gamification techniques (in the beginning) or gamification elements. I suggest that the term gamification be removed from the title and even from the keywords. Same applies for the hypotheses.

If the authors want to keep the concept of gamification they will have to develop the concept further and rework the design so that it is effectively implemented (see also comment 8). The paper available in https://scholarspace.manoa.hawaii.edu/items/d0812d74-4dd7-49e5-be6b-3fbd48262123 can be an excellent point to begin with, it has also a good reference list of other works that the authors can read.

However, from the description given, Coco-shapes can be considered a game, more specifically a serious game. As pointed out by Karl Kapp “A game is a system in which players engage in an abstract challenge, defined by rules, interactivity, and feedback, that results in a quantifiable outcome often eliciting an emotional reaction.” (Kapp, 2012, p.7)

The methodology also needs to be clarified and substantiated, see comments 10 and 11.

Revise the text because there are some longer sentences that become difficult to understand.

Comments:

1) Please clarify:

it was clear that no work approach is concerned with establishing a

34

precise and formally defined process to guide the design of a multimedia experience as a

35

fundamental part of the design stage of an interactive multimedia system.

2) Since the tool rating and comments are not mandatory for those using it, how can you infer what you mention? Please identify the source on which you base this statement:

In general, there are low

48

ratings and comments, implying that the solutions do not meet the user's expectations

49

and needs.

A theoretical review of the concepts is needed in the introduction, many of the claims are based on common sense ideas, which makes the research not very reliable!

4) The introduction should be divided into 3 sections: Introduction; Theoretical review (need major development) and Contextualization of the study. The choice of titles should be left to the authors' decision. This will simplify the reading. This 

5) Are there laws in Colombia that regulate the design of interactive multimedia experiences? Please clarify:

Once the regulations and laws that influence the design of the interactive multime-

115

dia experience were identified

 6)

(i) the basis of a relevant problem, in this

122

case the limitation of existing methodologies for the development of interactive and

123

multimedia systems, specifying a working approach for the design of interactive multi-

124

media experiences, (ii) a practice for the design of interactive multimedia experiences as

125

a solution artifact, (iii) the evaluation of the practice by means of a case study for the

126

school context, and (iv) a dissertation on the contribution and limitations observed as a

127

result of the evaluation of the practice for the design of interactive multimedia experi-

128

ences.

Suggestion for easy reading:

(i)               the basis of a relevant problem – in this case, the limitation of existing methodologies for the development of interactive and multimedia systems, specifying a working approach for the design of interactive multimedia experiences,

(ii)              a practice for the design of interactive multimedia experiences – a solution artifact,

(iii)            the evaluation of the practice – a case study for the school context, and

(iv)            a dissertation on the contribution and limitations – evaluation of the practice for the design of interactive multimedia experiences.

This will help those who read to distinguish the methodology from what are you implementing

7) Fig 1 is very difficult to read, it is possible to made it with larger letters?

8) 2.5 and table 5

From the text, it isn't easy to understand what the authors mean by gamification elements. Clarify how the techniques contribute to gamification and what gamification mechanics are intended to implement. Gamification elements are a vague concept and it is not understood what the authors mean.

9)

As part of the practice proposed in this study, the T6 technique suggests the defini-

600

tion of concrete gamification elements that contribute to the conception of a playful and

601

fun multimedia experience, so that students can enhance their skills.

Which gamification elements? Once again, no concrete gamification is mentioned! 

On T6 is mentioned: “It is advisable to consider elements such as [24]: goals and objectives, rules, narrative, freedom to choose, freedom to make mistakes, rewards, feedback,  visible status, cooperation and competition, time constraint, progress, and surprise.” How was it implemented on CocoShapes?

 10)

In the process of understanding the student's attitude

624

towards technology and learning outcomes, the findings during the evaluation reveal a

625

positive correlation; children are excited to know that the next level of an activity is more

626

difficult than the previous one. Similarly, children seem excited to see that the narrative

627

of the multimedia experience is based on an amusement park, has a character that is fa-

628

miliar to them, and includes a roller coaster and fairground rides, among other elements

629

that generate surprise for them.

How was this data collected and processed? It is only mentioned in the methodology that knowledge acquisition was measured. Was observation carried out, and which type?

11)

This sug-

643

gests that an interactive multimedia experience based on gamification, as is the case with

644

Coco-Shapes, favors learning results and increased knowledge, in the context of basic

645

education, and influences the attitude and reactions of the students.

How were other variables controlled? For this case, the teacher applied “methodology defined from his experience and using Coco-Shapes to accompany the activities associated with each theme.” (L: 496-498) How can you ensure that the difference is really gamification in the Coco-shapes and not the fact that it's another teacher with more experience, another type of methodology or children with better learning abilities? Or was it the fact that the children work better in the afternoon than in the morning?

You need to clarify the methodology used and how all the variables were controlled. How were the morning and afternoon children selected? Was it the same teacher? Was it already the usual teacher, or was it the first time they worked with that teacher? Was the observation team present in both situations or did they collect feedback only from the teacher in both situations or only in one? Had the children already worked on these concepts in their mother tongue? Are there differences in the acquisition of knowledge between the mother tongue and English?

Reviewer 2 Report

Dear authors,

I appreciate your manuscript. I appreciate the time and effort you put into producing this manuscript.

I would like to emphasize the following:

Major concern:

I am genuinely concerned about the clarity and organization of your manuscript. In my opinion, the primary issue is the absence of a clear and transparent contribution to the body of knowledge. Therefore, I strongly advise you to focus your research on theoretical or literary writing. This would clarify your contribution to the development of theory or knowledge.

Second, the structure of your manuscript is difficult to comprehend for the majority of your audience or readership. The manuscript is too long, and some of the subtopics lack a clear need based on the prevalent approach and writing style of prominent academic journals. Regarding this issue, I would suggest reorganising the manuscript and possibly hiring a professional academic editor to assist with the writing flow and idea delivery mechanism of the manuscript.

Comments:

Abstract – "basic education" must be defined succinctly to pique readers' interest. I would like to suggest that the authors emphasize the manuscript's contributions to the existing body of knowledge or theories.

2.3 - The hypotheses H1 and H2 are not supported by literature or empirical evidence. This is a typical academic writing standard, but I believe the authors need to make substantial revisions for this subtopic.

2.4 – Define "Essence practise." Perhaps the authors could provide additional information in the manuscript.

2.4 - The "two work routes" are ambiguous. Why is it essential? How does the implementation of "two work routes" improve the development process? As this is not a conventional approach, it would be intriguing to learn more about it.

Table 1. – What is T1, T2…. In the techniques?

Table 2. - It is preferable to present the code technique before Table 1. Not after Table 1

The remainder of the manuscript is overly complicated, which may reduce reader interest and comprehension, thereby jeopardising your chances of receiving citations.

Discussion – Absence of a crucial element, as the findings were not discussed exhaustively by drawing inferences and constructing arguments with the aid of literature.

Thank you very much.

Round 2

Reviewer 2 Report

I appreciate the changes made by the authors. The corrected version of the manuscript is now superior to the original.

Thank you.